# Transcriptome Analysis Reveals Modulation of Human Stem Cells from the Apical Papilla by Species Associated with Dental Root Canal Infection

**DOI:** 10.3390/ijms232214420

**Published:** 2022-11-20

**Authors:** Yelyzaveta Razghonova, Valeriia Zymovets, Philip Wadelius, Olena Rakhimova, Lokeshwaran Manoharan, Malin Brundin, Peyman Kelk, Nelly Romani Vestman

**Affiliations:** 1Department of Microbiology, Virology and Biotechnology, Mechnikov National University, 65000 Odesa, Ukraine; 2Department of Odontology, Umeå University, 90187 Umeå, Sweden; 3Department of Endodontics, Region of Västerbotten, 90189 Umeå, Sweden; 4National Bioinformatics Infrastructure Sweden (NBIS), Lund University, 22362 Lund, Sweden; 5Section for Anatomy, Department of Integrative Medical Biology (IMB), Umeå University, 90187 Umeå, Sweden; 6Wallenberg Centre for Molecular Medicine, Umeå University, 90187 Umeå, Sweden

**Keywords:** stem cells from the apical papilla (SCAP), regenerative endodontic treatment (RET), osteogenesis, dentinogenesis, *Enterococcus faecalis*, *Fusobacterium nucleatum*, transcriptome analysis, differential gene expression analysis (DEG)

## Abstract

Interaction of oral bacteria with stem cells from the apical papilla (SCAP) can negatively affect the success of regenerative endodontic treatment (RET). Through RNA-seq transcriptomic analysis, we studied the effect of the oral bacteria *Fusobacterium nucleatum* and *Enterococcus faecalis*, as well as their supernatants enriched by bacterial metabolites, on the osteo- and dentinogenic potential of SCAPs in vitro. We performed bulk RNA-seq, on the basis of which differential expression analysis (DEG) and gene ontology enrichment analysis (GO) were performed. DEG analysis showed that *E. faecalis* supernatant had the greatest effect on SCAPs, whereas *F. nucleatum* supernatant had the least effect (Tanimoto coefficient = 0.05). GO term enrichment analysis indicated that *F. nucleatum* upregulates the immune and inflammatory response of SCAPs, and *E. faecalis* suppresses cell proliferation and cell division processes. SCAP transcriptome profiles showed that under the influence of *E. faecalis* the upregulation of *VEGFA*, *Runx2*, and *TBX3* genes occurred, which may negatively affect the SCAP’s osteo- and odontogenic differentiation. *F. nucleatum* downregulates the expression of *WDR5* and *TBX2* and upregulates the expression of *TBX3* and *NFIL3* in SCAPs, the upregulation of which may be detrimental for SCAPs’ differentiation potential. In conclusion, the present study shows that in vitro, *F. nucleatum*, *E. faecalis*, and their metabolites are capable of up- or downregulating the expression of genes that are necessary for dentinogenic and osteogenic processes to varying degrees, which eventually may result in unsuccessful RET outcomes. Transposition to the clinical context merits some reservations, which should be approached with caution.

## 1. Introduction

The oral cavity is an ecological niche for a diverse range of microorganisms, including more than 700 bacterial species [1]. Microorganisms and their host live in a flexible balance that risks disturbance if traumatic dental injuries (TDIs) or oral diseases such as dental caries occur. Accordingly, if microorganisms penetrate dental structures and disturb host cell homeostasis, it can lead to infection (pulpal necrosis), periapical inflammation, and bone resorption (apical periodontitis) [2].

Pulpal necrosis in immature necrotic teeth demands challenging clinical procedures that pose a risk to teeth’s long-term survival. Conventional treatment of immature necrotic teeth includes apexification, which consists either of a long-term application of calcium hydroxide (CH) [3] paste to induce an apical barrier or placement of mineral trioxide aggregate (MTA) [4] to achieve closure of the root apex. These procedures, however, do not result in promoting root development [5] and may increase susceptibility to cervical fractures [6].

A currently promising therapeutic option—regenerative endodontic treatment (RET)—encourages tissue regeneration by promoting continued root development [7]. RETs are capable of regenerating vascular dental pulp tissue in animal models and human patients by ex vivo expanded autologous tooth stem cells implanted from deciduous teeth [8]. In this context, stem cells found at the apical papilla of the tooth (SCAP) are a unique group of dental stem cells characterized by their plasticity, potency, and versatility [9]. SCAPs have been shown to differentiate into osteo/odontoblast-like cells, adipocytes, and chondroblasts [10]. Moreover, SCAPs have been proven to take part in the processes of neurite outgrowth and axonal targeting, both in vitro and in vivo [11]. In comparison with dental pulp stem cells (DPSCs) and bone marrow mesenchymal stromal cells (BMMSCs), SCAPs show a higher proliferation rate and osteo/dentinogenic potential, and express multiple osteogenic markers including dentin sialophosphoprotein (DSPP), osteocalcin, and alkaline phosphatase (ALP) [12].

The presence of bacteria such as *Streptococcus oralis* and *Actinomyces naeslundii* and their residuals in the root canal, even after disinfection methods [13], can jeopardize RET success. In fact, micro-environmental conditions, such as pre-existing infection, seem to influence the viability, proliferation, and mineralization capacity of SCAPs [14,15]. Previous studies have confirmed that products from *S. oralis* J22 and *A. naeslundii* T14V-J1 inhibited mineralization of human SCAPs [13]. Moreover, lipopolysaccharides (LPS) extracted from *Porphyromonas gingivalis* resulted in a pronounced osteogenic response, since it significantly upregulates bone sialoprotein gene expression [16].

In a previous study, we reported that key species in dental root canal infection, namely *A. gerensceriae*, *S. exigua*, *F. nucleatum* and *E. faecalis* were able to modulate SCAPs under oxygen-free conditions in a species-dependent fashion. *Moreover*, *E. faecalis* and *F. nucleatum* reported the strongest binding capacity and significantly reduced SCAP proliferation [15]. As a diverse commensal and opportunistic bacterium, *F. nucleatum* participates in a variety of interactions with other bacteria and human cells, and these interactions can vary from helpful to damaging [17]. *F. nucleatum* is considered to be a key species in a biofilm formation, supporting primary colonizers such as Streptococcus species, and providing a low-oxygen microenvironment in the root canal, thereby protecting a secondary colonizer such as *P. gingivalis* [18]. *F. nucleatum* has been highly correlated with traumatized teeth [19], oral and extraoral human diseases such as periodontitis [20], endodontic infection [21], inflammatory bowel disease [22], and colorectal cancer [23]. Recently, it was shown that *F. nucleatum,* when exposed to immortalized primary colonic epithelium and vascular endothelial cells, upregulated genes related to inflammation, downregulated genes related to histone modification, and significantly remodeled chromatin states [24].

*Enterococcus faecalis* is associated with failed endodontic treatment [25] and is known for its survival capacity even in harsh conditions thanks to its virulence factors (enterococcal surface protein (esp), gelatinase (gelE), aggregation substance (asa1), cytolysin B (cylB) etc.) and ability to form biofilms [17,26]. It was reported that *E. faecalis* biofilm downregulated dentinogenic genes and upregulated osteoblastic genes in SCAPs [27]. *E. faecalis* is the most frequently isolated bacterial species from symptomatic root canal-treated teeth, accounting for up to 90% of cases [28]. Due to its ability to persist in harsh conditions with nutrient destitution and high alkalinity and despite the presence of intracanal medication, *E faecalis* is commonly found in secondary or chronic cases [29]. Forming a biofilm that is 1000-fold more impervious than planktonic bacteria to the action of anti-microbials, *E. faecalis* represents a particular pathogenicity and eradication problem [17]. Moreover, *E. faecalis* promotes the differentiation of murine bone marrow stem cells into CD11c-positive dendritic cells with aberrant immune functions while retaining the ability to induce proinflammatory cytokines [30].

Transcriptome profiling by RNA sequencing (RNA-seq) has been widely used to provide far higher coverage and greater resolution of the dynamic nature of the transcriptome. Unlike the genome (which is usually the same for all cells of the same lineage), the transcriptome can vary greatly depending on environmental conditions [31].

In the present study, we used RNA-seq transcriptomic analysis to reveal the alteration of gene expression across the SCAP genome in the case of *F. nucleatum* and *E. faecalis* stimulation, which lays a foundation for understanding the cellular changes induced by bacterial stimulation. Furthermore, we searched for osteogenic- and dentinogenic-associated genes expressed by SCAPs upon *F. nucleatum* and *E. faecalis* stimulation. 

## 2. Results

### 2.1. Mapping and Quantifying SCAP Transcriptomes by RNA-Seq

A total of 109.7 GB of raw sequence data was generated from all fifteen samples (Table 1). There were on average 49 million paired-end reads (2 × 150 bp) for each sample.

### 2.2. SCAP Transcriptome Elicits Distinct Profiles Based on Bacterial Stimulation

Principal component analysis (PCA) of the transcriptome profile of all 15 samples revealed four major clusters based on treatment variants. PCA is a dimensionality-reduction method, which helps to visualize differences between expressed gene profiles in controls and in treated samples, and to find the patterns in a dataset. Different clusters are formed by the transcriptomic profiles of SCAPs (donors I–III) treated by viable *E. faecalis* (planktonic stage), *E. faecalis* (supernatant), and viable *F. nucleatum* (planktonic stage). However, the unstimulated SCAP (negative control) and SCAPs co-cultured with *F. nucleatum* supernatants clustered together (Figure 1).

### 2.3. E. faecalis Supernatant Has the Strongest Influence on SCAPs Whereas Supernatant of F. nucleatum Has a Mild Effect

The transcriptome profile of uninfected versus infected SCAPs was compared using differentially expressed genes (DEG) analysis.

As a result of DEG analysis of transcriptomic profiles of SCAPs under the influence of bacteria or their supernatants, direct bacterial contact with *E. faecalis* resulted in differential expression of 1350 genes, and its supernatant resulted in the differential expression of 1453 genes. Moreover, 1252 genes were identified as DEGs via treatment with viable *F. nucleatum*, and only 135 genes by *F. nucleatum* supernatant infection (Figure 2).

Furthermore, differentially expressed genes (DEGs, *p* < 0.05 and log_2_ fold change > 1.5) were used to compare SCAP gene expression patterns by different treatment variants. For each comparison, the number of unique and shared genes was investigated, as shown in Figure 3. First, a comparison of up- and downregulated genes after SCAP co-cultivation with viable *F. nucleatum* and *E. faecalis* showed a similar number of co-upregulated genes; 191 genes in total. There were 207 genes which were commonly downregulated for direct bacterial treatment with *F. nucleatum* and *E. faecalis.* A comparison of bacterial supernatant treatments showed that the number of individually downregulated expressed genes was much higher under the influence of *E. faecalis* supernatant (808) than after *F. nucleatum* supernatant treatment (at 16). A difference in individually upregulated genes was also observed: 563 genes for *E. faecalis’s* supernatant and 37 genes for *F. nucleatum’s* supernatant. A higher number of differentially expressed genes (both up- or downregulated) would have a higher influence on the SCAPs. In this context, *E. faecalis’s* supernatant has the strongest influence on SCAPs. In contrast, *F. nucleatum’s* supernatant was characterized as having a very mild effect on SCAPs.

The Tanimoto coefficient was used to assess the degree of similarity between each compared pair of DEG profiles. A Tanimoto coefficient value of = 1.0 represents the highest degree of similarity between the two sets of elements. Comparing different pairs of DEG profiles, the highest Tanimoto coefficient, i.e., the greatest similarity between the two DEG profiles, was found for a pair of *E. faecalis* bacterial and *E. faecalis* supernatant treatments (T = 0.25). Accordingly, as the highest level of Tanimoto coefficient was 0.25 between treatment variants, it is suggested that the investigated treatments influence SCAPs co-equally.

### 2.4. F. nucleatum Upregulates Immune and Inflammatory Response whereas E. faecalis Downregulates Cell Division and Proliferation in SCAP

To investigate the function of the DEGs, gene ontology (GO) term enrichment analysis was performed using the Database for Annotation, Visualization and Integrated Discovery (DAVID). The GO analysis presented the top GO terms for upregulated and downregulated DEGs sorted by *p*-value and categorized into biological processes as a functional group (Figure 4).

Co-culture of SCAPs with *E. faecalis* (planktonic and supernatant) leads to activation/downregulation of several analogous processes in SCAPs (Figure 4A,B). In this context, the identical upregulated biological processes were responses to stress, positive regulation of apoptotic processes, and oxidation-reduction processes (Figure 4A). Cell division, cell cycle regulation, cell proliferation, and regulation of signal transduction by p53 were commonly downregulated biological processes in SCAPs co-cultured with viable *E. faecalis* and its supernatant (Figure 4A,B).

Similarly, a set of identical biological processes was upregulated/downregulated in SCAPs co-cultured with viable *F. nucleatum* and *F. nucleatum* supernatants. The identical upregulated processes were apoptotic processes, inflammatory responses, cellular response to lipopolysaccharides, and immune responses; whereas mitotic nuclear division was the only commonly downregulated biological process (Figure 4C,D). It is worth mentioning that despite the common regulated biological processes between direct and indirect bacterial contact, there was both up- and downregulation of biological processes that were unique to each type of treatment. For example, only the *F. nucleatum* supernatant led to the upregulation of the negative regulation of cell proliferation and downregulation of cytokinesis processes (Figure 4D). Conversely, the supernatant of *E. faecalis* upregulated the processes of positive regulation of cell proliferation, the cellular response to lipopolysaccharides, the cellular response to hypoxia, etc.—processes that were not stimulated under the direct *E. faecalis* stimuli (Figure 4B). Overall, *F. nucleatum* modulated SCAP through upregulated genes which were mainly involved in immune and inflammatory responses, and downregulated genes mainly involved in the process of cell cycle regulation. Interestingly, the effect of *F. nucleatum* supernatants on SCAPs was very mild and was evidenced by the small gene count, as an exception among other treatment cases.

### 2.5. Osteogenic/Odontogenic Genes in SCAPs Are Strongly Influenced by F. nucleatum and E. faecalis Associated with Endodontic Infections

Several genes were chosen for their role in the mineralization process and were sorted into the following categories: dentinogenic, osteogenic cell surface, or osteogenic intracellular and osteogenic secreted genes (Table 2). Accordingly, Figure 5 showed SCAP transcriptomic analysis and corresponding genes that were up- or downregulated.

The largest number of dentinogenic/osteogenic-associated genes were regulated when SCAPs were exposed to *E. faecalis* supernatants. Vascular endothelial growth factor A (*VEGFA*), fibroblast growth factors (*FGF2*), runt-related transcription factor 2 (*RUNX2*), T-box transcription factor 3 (*TBX3*), and nuclear factor interleukin 3 (*NFIL3*) were upregulated in comparison with non-treated SCAPs. In contrast, the parathyroid hormone 1 receptor (*PTH1R*); WD repeat-containing protein 5 (*WDR5*); signal peptides; CUB domain and EGF-like domain (Epidermal growth factor) containing 3 (*SCUBE3*); insulin-like growth factor-binding protein 3 (*IGFBP3*); and collagen, type I, alpha 1 (*COL1A1*) were downregulated in comparison with non-treated SCAPs.

Co-culture of SCAPs with viable *E. faecalis* led to the regulation of five osteogenic genes. *NFIL3*, the T cell immune regulator gene 1 (*TCIRG1*), and gamma-aminobutyric acid B receptor 1 (*GABABR1*) were upregulated, and *WDR5* and *IGFBP3* were downregulated. In addition, five dentinogenic/osteogenic genes were regulated when SCAPs were exposed to *F. nucleatum*. *VEGFA*, *TBX3*, and *NFIL3* were upregulated, and *WDR5*, as well as T-box transcription factor 2 (*TBX2*), were downregulated. Only one osteogenic-associated gene was upregulated when SCAP was co-cultured with *F. nucleatum* supernatant (*NFIL3*). Interestingly, *NFIL3* was upregulated in all treatment variants (Figure 5).

## 3. Discussion

A transcriptome is the whole set of cell transcripts, and their quantity for a specific stage of cell development or physiological state. Comprehension of cell transcripts as transcriptomes is critical for development and disease conception, as well as for elucidation of the functional aspects of the genome and in disclosing the molecular elements of cells and tissues [72]. Through RNA-seq transcriptomic analysis, we managed to obtain transcriptomic profiles of healthy SCAPs under the exposure of opportunistic bacteria of the oral cavity and their metabolites. In this study, with the help of transcriptomic analysis, we established that the *E. faecalis* supernatant showed a distinguished stimulant effect on the expression of osteo- and dentinogenic genes in SCAPs, compared with other treatment types. In contrast, cells treated by *F. nucleatum* supernatants had the smallest effect on the expression of these genes. This work on a cellular model of healthy SCAP modulated by oral bacteria in vitro is an initial step towards understanding the complex processes potentially occurring in the in vivo ecosystem of SCAP and oral bacteria.

Moreover, PCA analysis of the transcriptome profiles of SCAPs formed four clusters founded by treatment variants. Three of these clusters were formed strictly by the treatment variant: SCAPs under the direct bacterial influence of *F. nucleatum*; SCAPs treated with the planktonic stage of *E. faecalis*; and SCAPs treated with *E. faecalis* supernatants. However, it is noteworthy that SCAPs treated by *F. nucleatum* supernatant (donors I–II) and untreated SCAPs (donors I–III) were grouped in one cluster.

In addition, differential gene expression analysis (DEG) has shown similarities between up- and downregulated genes in *E. faecalis* and *F. nucleatum* in direct bacterial treatment. However, when comparing the action of the supernatants of these bacteria, the number of DEGs under the influence of the supernatant of *E. faecalis* was ten times higher than DEGs under the influence of *F. nucleatum* supernatants. The similarity between the two types of cell treatment was evaluated using the Tanimoto coefficient, which in this case was the smallest (0.05). The low value of Tanimoto coefficient (0.06) in comparing direct and indirect bacterial contact by DEG number confirmed the meager influence of *F. nucleatum* supernatants on SCAP. The strength of the treatment’s influence on stem cells from the strongest to the weakest was as follows: *E. faecalis* supernatant > *E. faecalis* bacteria > *F. nucleatum* bacteria > *F. nucleatum* supernatant.

According to the search of gene ontology annotation, in between upregulated biological processes, apoptotic processes were upregulated in all treatment variants: through both direct contact with bacteria in the planktonic stage and through supernatants saturated by bacterial metabolites. This means that the presence of bacteria in the planktonic stage or secreted bacterial metabolites in the tooth at the investigated concentrations can possibly lead to root aberration and failure of regeneration treatments, as was already shown in several studies using other cells or models [73,74]. The key differences between the influence of *F. nucleatum* (bacteria as well as supernatant) in comparison with *E. faecalis* (bacteria as well as supernatant) on SCAPs were the upregulation of the immune and inflammatory responses occurring only in the case of SCAP treatment with *F. nucleatum* bacteria/supernatant. Intensification of inflammatory processes would lead to dentist attention and improve the probability of improving the regenerative process [75], whereas the effect of *E. faecalis* bacteria in either the planktonic stage or bacterial metabolites may not be detected as quickly because it remains hidden from the immune response [76]. Chong et al. demonstrated this on a mouse model with non-healing wounds infected with *E. faecalis* that led to the suppression of the inflammatory cytokines, notwithstanding the immune cell infiltration [77]. It was shown that the ability of bacteria to reduce proinflammatory cytokine secretion is an intrinsic feature that reflects the degree of bacterial virulence [78].

Interestingly, the number of gene counts in the biological processes that were identified via the gene ontology tool were shifted to the side of downregulated processes in the case of SCAP co-culture with *E. faecalis*. If SCAPs were cultured in *E. faecalis* supernatant, biological processes were balanced between up- and downregulation. Upregulated biological processes predominated if SCAPs were treated by *F. nucleatum* bacteria or its supernatant. The decreased number of gene counts for upregulated genes could be an indication of cell infection with pathogenic bacteria, as was reported by Stekel and co-authors on a model of human intestinal epithelial cells [79].

It is noteworthy that the regulated biological processes found via the GO tool for SCAPs co-cultured with bacteria and for SCAPs cultured in the supernatant from the corresponding bacteria were not only similar, but also unique in their upregulated and downregulated processes. This very likely indicates that the effect of bacteria on cells is mediated not only by direct contact with microbial-associated molecular patterns (MAMP) [80], but also, possibly, by secreted bacterial metabolites [81].

Recent findings considered SCAPs to be a new and promising source of stem cells for regenerative endodontic treatments [9,82,83] which is why the key issue in the present study was to examine the influence of two specified bacterial species, isolated from the root canal, on the differentiation directions of SCAPs. To this end, 33 dentinogenic- and osteogenic-related genes that were selected according to their role in mineralization processes (Table 2) were examined in the transcriptomics/gene expression profile of non-infected SCAPs, after which the expression of these genes was compared with gene expression profiles in the treated groups. The selected genes were divided into two subgroups according to the genesis of either dentin or bone (osteogenic or dentinogenic markers), and the subgroup of osteogenic markers was further divided according to the marker’s location in the cell: on the cell surface, intracellular, or secreted.

This study showed that direct bacterial treatment with SCAPs and the bacterial metabolites of *F. nucleatum* affect the expression of dentinogenic and osteogenic intracellular-associated genes in various ways. One important gene for dental pulp repair and reparative dentine formation—*VEGFA*—was upregulated only in the group that underwent direct bacterial treatment with *F. nucleatum*. Mendes et al. showed that *F. nucleatum* was able to induce an immune response in endothelial cells and increase *VEGF* cell secretion, but the expression of this gene at the mRNA level was lower [84]. Immunohistochemical data of cells from the inflammatory infiltrate of irreversible pulpitis showed strong positive *VEGF* expression [85]. Despite the fact that *VEGF* is important for pulp healing and angiogenesis in general [86], stimulation of the expression of this gene by *F. nucleatum* may indicate an inflammatory process in SCAPs, since angiogenesis may in fact increase the severity of the inflammatory process [85].

We have furthermore shown that direct bacterial contact of SCAPs with *F. nucleatum* leads to upregulated expression of two other osteogenic intracellular markers: *TBX3* and *NFIL3*. A previous study conducted by Govoni et al. had demonstrated that *TBX3* inhibits mineralization of osteoblast cells on mouse pre-osteoblast cells [53]. Another upregulated gene—*NFIL3*—was found to be upregulated in all variants of SCAP treatment and the only osteogenic gene that was regulated in the case of SCAP culture in *F. nucleatum* supernatant. Besides the involvement of *NFIL3* in the development of the innate immunity cells in a mouse model [87], it has been shown that *NFIL3* can act as a transcriptional repressor in osteoblasts while exhibiting differential activity as an activator in osteocytes on mouse osteoblastic cell lines [49].

However, direct bacterial contact of *F. nucleatum* with SCAPs resulted in the downregulation of two genes, *WDR5* and *TBX2*. It should be noted that downregulation of the expression of these genes was also observed under direct and indirect bacterial exposure to *E. faecalis*. Studies show that the expression of these genes works as a stimulator of the canonical Wnt pathway, and therefore acts as a stimulus for differentiation of odonto- and osteoblasts [52,88].

Concerning the other treatment group, *E. faecalis’s* bacterial suspension and supernatant, it was shown that the supernatant of these bacteria modulated SCAPs to a greater extent. Direct bacterial stimulation of SCAPs with *E. faecalis* resulted in the upregulation of three genes: *GABABR1*, *TCIRG1* and *NFIL3*. To date, there is no data on how the expression of these genes affects the osteo/odontogenic potential of SCAP, but it is known that the secretion of these factors may indicate an inflammatory process and inhibition of osteoblastogenesis [43,49,89].

Interestingly, it was the indirect effect of *E. faecalis* on SCAPs (via the bacterial supernatant) that resulted in the greatest influence on the transcriptomic SCAP profile. Only in the case of indirect bacterial contact was the expression of *VEGF* and *FGF2* observed. As mentioned before, *VEGF* is important both in angiogenesis and osteogenesis; however, recent studies show that overexpression of *VEGF* by dental pulp cells exposed to gram-positive bacterial toxins may, in the long run, lead to pulp necrosis owing to intra-pulpal pressure caused by *VEGF* [90]. The upregulated expression of *RUNX2*, *TBX3*, and *NFIL3* indicate that under the influence of *E. faecalis*, SCAPs’ differentiation potential may be affected negatively [91]. We strongly emphasize the observation that only this treatment option caused the upregulation of *RUNX2*. *RUNX2* could be activated by the mitogen-activated protein kinase (MAPK) pathway, and in turn this pathway could be stimulated by cell treatment with the osteogenic growth factor, FGF2 [92]. Since in the present study *FGF2* expression was found to be upregulated only in the *E. faecalis* supernatant treatment option, we can assume that *RUNX2* is being activated and it can induce differentiation to osteoblasts instead of odontoblasts [91]. In addition, with *E. faecalis* supernatant treatment, the downregulation of *PTH1R*, *SCUBE3*, *COL1A1m,* which are important for osteogenic differentiation, was shown [57,93,94,95]. Furthermore, it is worth noting that the expression of the *IGFBP3* gene was downregulated in this type of SCAP treatment, and a recent study by Aizawa et al. in a mouse model showed that *IGFBP3* is required for pre-odontoblast differentiation [96].

Additionally, the transcriptomic analysis is very specific, sensitive, and reproducible, and allows for the assay of multiple samples simultaneously [72].

SCAPs are a promising source of local stem cells, which could be used for regenerative endodontic treatment (RET) of traumatic dental injuries to immature teeth [97]. Traumatic dental injuries create the possibility for oral bacteria, which exist in the planktonic stage, to come into direct contact with SCAP cells [98] through the surface components of bacteria (flagella, pili, surface layer proteins, capsular polysaccharides, lipoteichoic acid, lipopolysaccharides) via SCAPs’ toll-like receptors (TLR). This then regulates several signaling pathways, such as nuclear factor B (NF-κB) and mitogen-activated protein kinases (MAPK), as was shown on epithelial cells [99,100]. The dysregulation of the latter pathway leads to diminishment of the mineralization processes in human dental pulp stem cells (DPSCs) [101,102].

However, oral bacteria exist in most cases in the oral cavity in the form of biofilms and can continuously secrete metabolites produced by bacteria, such as vesicles, extracellular proteins, organic acids, indoles, etc. Those metabolites can easily reach the SCAPs in the case of traumatic dental injury and, probably, stimulate signaling cascades and the production of specific cytokines and chemokines, which then up- or downregulate inflammation, resulting in a changed microenvironment which can influence regeneration [27,103,104,105]. The success of regenerative endodontic treatment depends on a better understanding of the consequences of SCAPs’ and live bacteria’s direct interaction in comparison with SCAP stimulation by bacterial metabolites alone.

Our results presume rigorous disinfection procedures in the case of traumatic dental injuries to be one of the keys to successful regenerative treatment. One limitation of this study is the short duration of cell exposure to different treatment options (with a longer exposure, the processes of cell death began to prevail due to live bacteria), which prevented us from seeing the development and the result of the differentiation process.

Undoubtedly, the influence on healthy SCAPs by bacteria, which are most often isolated from infected root canals, is improperly perceived as a model of an infected injured tooth; however, interest in and feasibility of the total elimination of the two proposed opportunistic bacteria merits further discussion on the one hand, and on the other hand so do the interactions of these two bacteria with other species of the oral microbiota. Thus, from a translational perspective, our investigations suggest that further studies investigating not only one species but a cell–microbiota interaction may be warranted in order to mimic the complex clinical situation.

Furthermore, extrapolating the results to mimic the situation of a chronic process should be done with care, as we are using an in vitro system. Nevertheless, our method still provides useful information to increase understanding of in vivo processes.

In summary, the differentiation of tissue-specific MSCs, such as SCAPs to osteoblasts, is a very complex process that is finely orchestrated by a number of cytokines, chemokines, signaling molecules, and mechanical stimuli. Such a stimulus should be applied at specific time points of the differentiation process [106,107]. In the previous study, we showed that *F. nucleatum* stimulates the inflammatory response of SCAP, whereas *E. faecalis* reduces it and decreases the level of the key proteins of Wnt/β-Catenin and NF-κB signaling pathways that are important for bone formation [98]. In the present study, stem cells were exposed to bacteria at the planktonic stage or to the corresponding bacterial supernatant over a 24 h period in vitro. Direct and indirect bacterial stimulation of cells led to a change in the transcriptomic profile of SCAPs, namely up- and downregulation of genes, changes in the levels of which may adversely affect the processes of osteo- and dentinogenesis.

## 4. Materials and Methods

### 4.1. Cell Isolation and Culture

In this study, we used three clinical isolates obtained from impacted human teeth (*n* = 3; two lower jaw third molars and one upper jaw canine) from three healthy patients (one male and two females with mean age of 17 years and a range 11–20 years), due to retention and/or lack of space in orthodontic treatment [10,108]. The authenticity of the multipotent stromal cells was confirmed by the presence of CD73, CD90, CD105, and CD146, and the absence of CD11b, CD19, CD34, CD45, and HLA-DR. Detection was performed with PE-conjugated antibodies against the above-mentioned markers by flow cytometry (FCM, Becton Dickinson, Accuri C6), and analyzed with FlowJo Software V9. The multipotency and stemness of isolated SCAPs from these donors, in a step toward adipogenic and osteogenic differentiation, were previously published [15]. Collection, culture, storage, and usage of all cell lines were approved by the local research ethics committee at Umeå University (Reg. no. 2013-276-31M).

The SCAPs we used in this study were isolated 4–5 years ago and cryo-preserved in cryomedium (90% FBS and 10% DMSO). SCAPs were brought back from cryopreservation and grown until 95% confluency in cell culture medium α-MEM, GlutaMAX™ GIBCO (ThermoFisher (Lifetech), Waltham, MA, USA, #32561029), supplemented by a 10% FBS and 1% Penicillin-Streptomycin solution (Merck (Sigma-Aldrich, St. Louis, MO, USA) # P0781). Cells were harvested using trypsin/EDTA solution (Merck (Sigma-Aldrich) #T3924), counted by Countess II Automated Cell Counters (ThermoFisher Scientific) according to the manufacture’s protocol, and seeded at a cell density of 2 × 10^4^ cells/mL into 10 cm cell culture dishes, and kept in the cell incubator at +37 °C with 5% CO_2_ overnight until adherence. SCAP cells (4th passage) were used for the experiments.

### 4.2. Bacterial Strains and Culture Conditions

We used clinical isolates of *F. nucleatum subsp. polymorphum* and *E. faecalis* (Appendix A) obtained from root canal samples of traumatized necrotic teeth of young patients that were referred to the Endodontic Department, Region Västerbotten, Sweden (Reg. no. 2016/520-31) in this study. Sample collection, processing, and characterization of isolates was performed as previously described [19]. Briefly, samples were collected from root canals under strict aseptic conditions. Before entering the pulp space, a rubber dam was applied and the tooth, clamp, and dam were disinfected with hydrogen peroxide (30%) and tincture of iodine (5%). The contents of the root canal were absorbed into sterile paper points. The paper points were then moved to a TE buffer followed by culturing in anaerobic conditions on Fastidious Anaerobe Agar (FAA) (Svenska LabFab) for one week. Colonies with different phenotypic patterns were selected from each plate, amplified by PCR, and sequenced to identify bacterial species [13]. Sequences were compared with the eHOMD database (Expanded Human Oral Microbiome Database, Forsyth, (http://www.ehomd.org (accessed on 9 February 2017) for detection at the species level with >98.5% sequence similarity with regard to their 16S rRNA genes.

The cryostocks of the designated species were preserved in 20% sterile skimmed milk and stored at −80 °C until needed for experiments. Bacteria were removed from the cryostocks and passaged on Fastidious Anaerobe Agar (FAA) (Svenska LabFab, Söderhamn, Sweden) medium supplemented with 5% citrated bovine blood (Svenska LabFab FIE 34200500) and 16 µg/L of vitamin K (Sigma-Aldrich M5850) in an anaerobic atmosphere (10% CO_2_, 10% H_2_, 80% N_2_) at +37 °C for 5–7 days.

Bacteria were then harvested and resuspended in cell culture medium MEM-α enriched with 10% FBS. Optical density of each bacterial suspension was adjusted by spectrophotometer at 600 nm to 1.0 (corresponds to 1 × 10^8^ CFU/mL). Bacterial suspensions were used directly in the co-culture experiment with SCAPs or for preparation of bacterial supernatants. Bacterial strains were quantitatively inoculated in MEM-α supplemented with 10% FBS in order to have the equivalent of multiplicity of infection (MOI) equal to 100. Each strain was grown individually over 24 h in anaerobic conditions at +37 °C. Bacteria were then pelleted by centrifugation at 10.000× *g* for 10 min at 4 °C and supernatants enriched by bacterial metabolites (hereafter referred to as ‘bacterial supernatants’) were filtrated through a sterile syringe filter with 0.22 µm pore size (Fisher Scientific #10268401), aliquoted, and stored at −80 °C until use.

### 4.3. SCAP Infection by Bacterial Strains: Co-culture Experiments

Species of *E. faecalis* and *F. nucleatum* were used in this study because of their strong binding capacity and proliferation effects on SCAPs [15]. Viable bacteria in the planktonic stage and bacterial supernatants were analyzed in the following treatment variants: (i) SCAPs (donors I–III) co-cultured with viable *F. nucleatum* (planktonic stage); (ii) SCAPs (donors I–III) co-cultured with viable *E. faecalis* (planktonic stage); (iii) SCAPs (donors I–III) co-cultured with *F. nucleatum* (supernatant); (iv) SCAPs (donors I–III) co-cultured with *E. faecalis* (supernatant); and (iv) SCAPs (donors I–III) without bacterial infection, used as controls.

Previous study showed that selected bacterial species modulated SCAP cytokine secretion only after 24 h of co-cultivation [15]. Based on the results of the previous study, SCAPs were cultured in an anaerobic atmosphere (10% CO_2_, 10% H_2_, 80% N_2_) at +37 °C for 24 h. For the co-culture experiments, viable bacteria or their supernatants were resuspended in antibiotic-free cell culture medium and adjusted to MOI 100 on SCAPs as previously described [15]. MOI 100 was determined to be an effective concentration of bacteria per cell, by both the dose response test and the neutral red cytotoxicity test [15].

### 4.4. RNA Extraction and Quality Evaluation

After bacteria or supernatants were co-cultured with SCAP for 24 h, cell monolayers were washed with PBS and detached via a trypsin/EDTA solution. Collected cells were treated using an RNA stabilizer (RNAprotect Cell Reagent Qiagen, #76526) and kept overnight at +4 °C. The next day, collected cell samples were lysed and homogenized (QIAshreder Qiagen, #79654). RNA extraction was performed using a RNeasy Mini Kit (RNeasy Mini Kit, Qiagen, #74104) according to the manufacturer’s protocol with an additional step of on-column treatment by DNase for elimination of residual DNA contamination (DNase I, RNase-free, ThermoFisher Scientific #EN0521). The quality of isolated RNA (yield, purity, and integrity) was assessed using an Agilent 2100 instrument (Agilent Technologies, Santa Clara, CA, USA).

mRNA extraction, conversion to complementary DNA (cDNA), and sequencing library preparation (150 nucleotides) were performed by Novogene Bioinformatics Technology Co., Ltd. (Beijing, China). The RNA integrity number (RIN)—an important tool in conducting valid gene expression measurement experiments—satisfied the quality requirements for transcriptomic analysis (mean 9.13) (Table 3). Thus, good RNA quality assessment is considered one of the most critical elements in obtaining meaningful gene expression data via transcriptomics [109]. Agarose gel electrophoresis confirmed that none of the samples were contaminated by DNA or protein and that the RNA was intact and showed two sharp 28S and 18S rRNA bands (Appendix A).

### 4.5. Transcriptomic Analysis, Data Preprocessing and Bioinformatics

The paired-end reads obtained from NovaSeq were checked for quality using FastQC [110]. Initially, we pre-processed the RNA-seq data from our fifteen samples using Cutadapt (v3.1) [111] by trimming reads containing adapter and poly-N sequences and low-quality raw data reads. All downstream analyses were based on clean data of high quality. The trimmed reads were aligned separately to the human genome by HISAT2 (v2.2.1) [112] using default parameters. The genome sequences and the human annotations (GRCh38.p13) were obtained from the NCBI genome database (https://www.ncbi.nlm.nih.gov/genome, accessed on 5 March 2021). Post-alignment QC metrics were generated using RSeQC (v2.6.4) [113], which detected junctions, read distribution, experiment type, and ribosomal contamination [114].

For quantitation of the mapped read numbers of each gene we used FeatureCounts (subread v2.0.0). Differential expression analysis was performed using DESeq2 (v1.32) [115]. Differentially expressed genes were those whose expression changed at least 1.5 times at the log_2_ level compared with the expression of genes in the control group, and those who had a level of statistical significance of *p* < 0.05. For clarity, DEGs exhibiting higher levels of expression in treated samples compared with controls were designated as ‘highly regulated’, whereas those exhibiting the opposite ratio were designated as ‘reduced’. Unwanted variation was removed using the removeBatchEffects function from Limma (v3.48.3) through technical heterogeneity [116]. DESeq2 was used to compute a VST (variance stabilizing transformation) of the original count data for visualization in principal component analysis (PCA) using R (4.0.3). Variability due to the three replicate data sets being higher (Appendix A), referred to as ‘batch effects’, was corrected using the ‘Limma’ R package [116] prior to further analysis to avoid introducing biologically irrelevant signals into the high-throughput data and misleading conclusions [117].

The Database for Annotation, Visualization and Integrated Discovery (DAVID; version 6.8; david.ncifcrf.gov/ (accessed on 20 April 2021)) was used to perform GO (www.geneontology.org (accessed on 20 April 2021)) enrichment; *p* < 0.05 was considered to indicate a statistically significant difference [118,119,120].

A Tanimoto coefficient was used to assess the degree of similarity between each compared pair of DEG profiles. A Tanimoto coefficient ranges from 0 (no similarity) to 1 (high similarity), where values greater than 0.85 reflect a high probability of similarity between the two sets of elements [121].

## Figures and Tables

**Figure 1 ijms-23-14420-f001:**
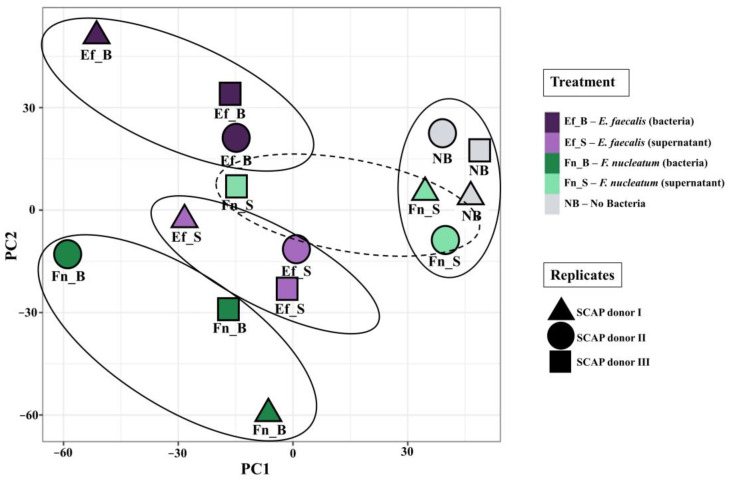
Principal component analysis of various transcriptomic profiles of SCAPs under different treatment conditions. SCAP donors are marked by shapes, and treatment variants are denoted by colors. Suggested clusters are circled. The first principal component (PC1) shown on the *x*-axis represents the most variation in the data, and the second principal component (PC2) on the *y*-axis represents the second-highest level of data variation. The axes are ranked in order of importance: differences along the PC1 axis are more important than differences along the PC2 axis.

**Figure 2 ijms-23-14420-f002:**
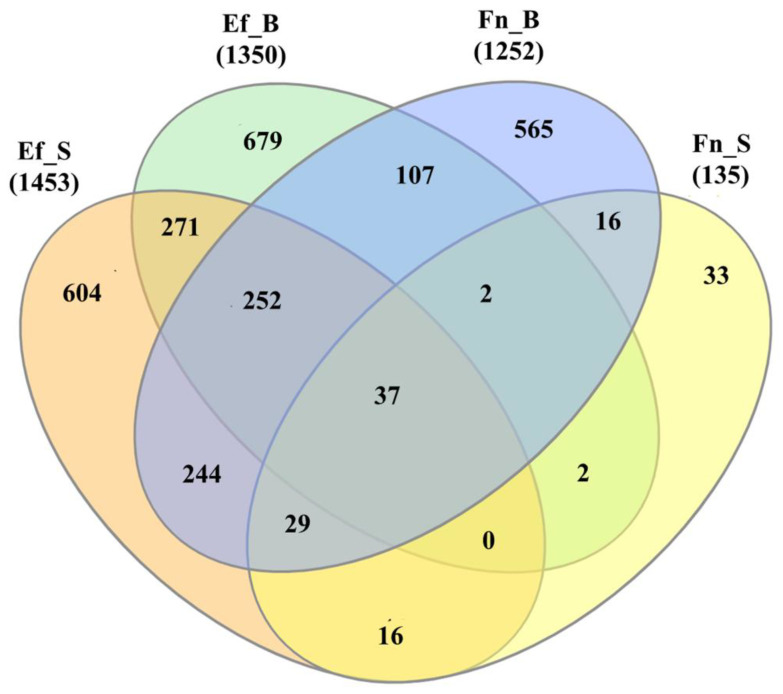
Venn diagram of SCAPs’ differentially expressed genes, by treatment variant. Data is shown in absolute numbers of genes, where the differentially expressed genes (in relation to control) in each treatment are compared with each other. Pooled data for SCAP (donors I–III): F.n_B: SCAP co-cultured with *F. nucleatum*; F.n._S: SCAP co-cultured with *F. nucleatum* supernatant; E.f._B: SCAP co-cultured with *E. faecalis*; E.f._S: SCAP co-cultured with *E. faecalis* supernatant.

**Figure 3 ijms-23-14420-f003:**
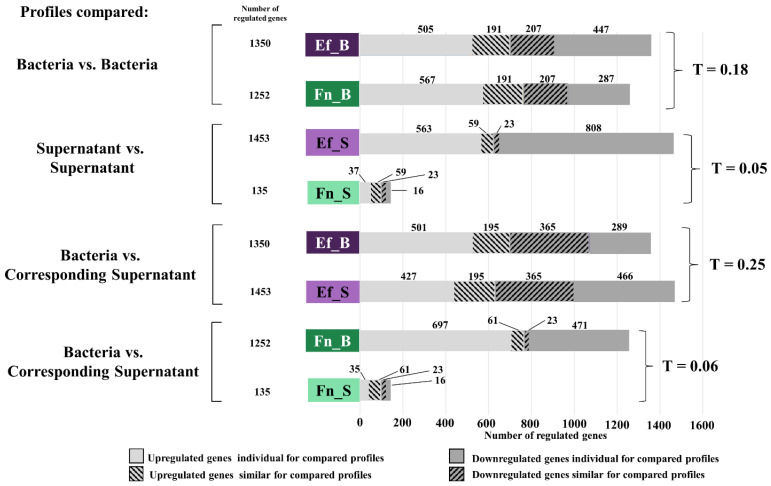
Bar graph of pairwise comparison of differentially expressed genes (DEGs) in absolute numbers. F.n_B: SCAP co-cultured with *F. nucleatum*; F.n._S: SCAP cultured with *F. nucleatum* supernatant; E.f._B: SCAP co-cultured with *E. faecalis*; E.f._S: SCAP cultured with *E. faecalis* supernatant. DEG degree of similarity was analyzed based on the Tanimoto coefficient (T; highest possible similarity = 1.0). The genes that were up- or downregulated exceptionally for one of the compared profiles are named as individual genes; shared genes that were up- or downregulated in both types of treatment are named as similar genes.

**Figure 4 ijms-23-14420-f004:**
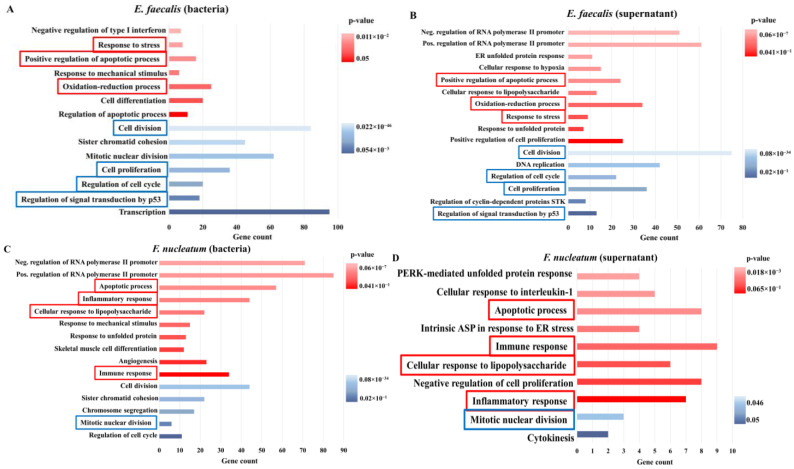
Gene ontology annotation. Upregulated biological processes (red bars) and downregulated processes (blue bars) sorted by *p*-value. Biological processes with the highest *p*-value are placed on the bottom, with the smallest *p*-value on the top (for both up- and downregulated processes): (**A**) SCAPs co-cultured with viable *E. faecalis*; (**B**) SCAPs co-cultured with *E. faecalis* supernatant; (**C**) SCAPs co-cultured with viable *F. nucleatum*; and (**D**) SCAPs co-cultured with *F. nucleatum* supernatants. Biological processes (upregulated or downregulated) that were similar between treatment with planktonic bacteria and corresponding supernatants are marked by red or blue frames, respectively. ER: endoplasmic reticulum, STK: serine/threonine kinase, PERK: PKR-like ER kinase, ASP: apoptotic signaling pathway.

**Figure 5 ijms-23-14420-f005:**
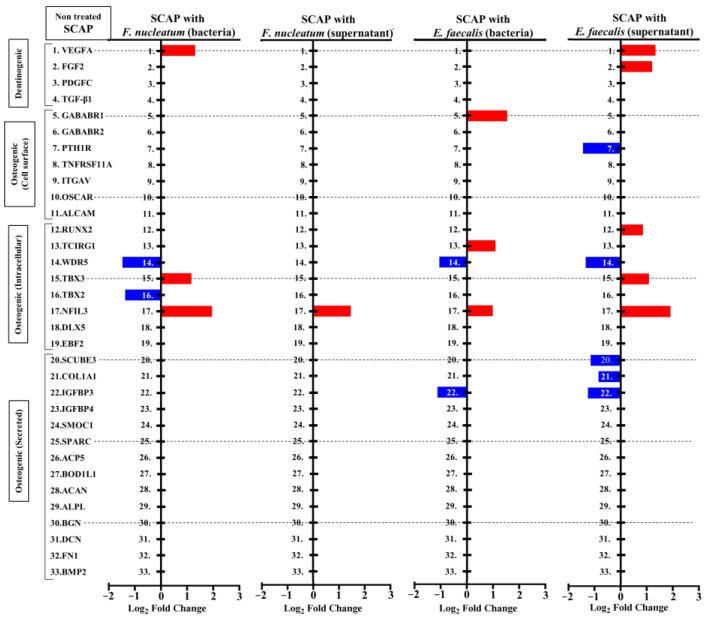
Up- and downregulation of genes associated with bone and dentin formation detected in SCAP culture variants. Transcriptome analysis of SCAP mRNA from three donors was performed. Gene expression is shown in log_2_ fold change. Blue boxes represent downregulation. Red boxes represent upregulation.

**Table 1 ijms-23-14420-t001:** RNA-seq data statistics. DI—SCAP donor I; DII—SCAP donor II; DIII SCAP donor III; NB—non-treated SCAP; F.n_B—SCAP co-cultured with *F. nucleatum;* F.n._S—SCAP cultured with *F. nucleatum* supernatant; E.f._B—SCAP co-cultured with *E. faecalis*; E.f._S*—*SCAP cultured with *E. faecalis* supernatant.

Sample	Type of Treatment	Raw Data (Read Count)	Clean Data (Read Count)	Total Mapped Reads	Total Mapping Rate (%)	Percent of Genome Regions (%)		
						Exon	Intron	Intergenic
1	DI NB	43,633,856	43,606,890	43,210,852	99.1	93.0	2.5	4.5
2	DI F.n_B	54,476,736	54,444,824	53,964,109	99.1	93.3	2.4	4.3
3	DI F.n._S	42,956,122	42,933,526	42,576,137	99.2	93.7	2.0	4.3
4	DI E.f_B	55,629,374	55,600,608	54,824,199	98.6	88.9	5.9	5.2
5	DI E.f_S	55,430,062	55,401,924	54,885,257	99.1	92.3	3.2	4.5
6	DII NB	47,027,536	47,002,190	46,541,473	99.0	93.5	2.2	4.3
7	DII F.n_B	48,806,760	48,784,160	48,246,591	98.9	91.6	2.9	5.5
8	DII F.n._S	46,427,206	46,406,140	45,986,861	99.1	92.1	2.7	5.2
9	DII E.f_B	39,262,262	39,242,992	38,806,871	98.9	90.9	4.2	4.9
10	DII E.f_S	49,064,034	49,042,722	48,616,751	99.1	91.9	3.3	4.8
11	DIII NB	48,797,068	48,773,676	48,361,861	99.1	93.7	2.1	4.2
12	DIII F.n_B	52,910,564	52,883,784	52,349,588	98.9	93.2	2.5	4.3
13	DIII F.n._S	52,127,896	52,105,306	51,686,329	99.2	91.9	2.7	5.4
14	DIII E.f_B	46,191,244	46,171,130	45,806,664	99.2	90.8	4.2	5.0
15	DIII E.f_S	49,193,684	49,173,978	48,823,399	99.3	93.4	2.3	4.3

**Table 2 ijms-23-14420-t002:** Genes which were examined in the SCAP transcriptome, sorted by their function in osteogenesis and dentinogenesis.

Gene		Function	Reference
**Dentinogenic genes**	Vascular endothelial growth factor A (*VEGFA*)	Inducing proliferation and differentiation of hDPSCs into odontoblasts	Matsushita et al., 2000 [32]
	Fibroblast growth factor 2 (*FGF2*)	Potent regulator of mineralization	Roberts-Clark and Smith, 2000; Madan and Kramer, 2005; Cooper et al., 2010; Miraoui and Marie, 2010; Marie et al., 2012; Smith et al., 2012 [33,34,35,36,37,38]
	Platelet-derived growth factor C (*PDGFC*)	Enhancement of DPSCs proliferation, odontoblast differentiation, and regeneration of dentin–pulp complex	Tsutsui, 2020 [39]
	Transforming growth factor β-1 (*TGF-β1*)	Importance in regulating reparative dentinogenesis	Toyono et al., 1997; Piatelli et al., 2004; Unterbrink et al., 2002 [40,41,42]
**Osteogenic cell surface markers**	Gamma-aminobutyric acid B receptor 1 (*GABABR1*)	Negative regulation of osteoblastogenesis	Takahata et al., 2011 [43]
	Gamma-aminobutyric acid B receptor 2 (*GABABR2*)		
	Parathyroid hormone 1 receptor (*PTH1R*)	Committing MSCs to the osteoblast lineage and promoting bone formation	Yu et al., 2012 [44]
	Receptor activator of nuclear factor-κB (*RANK*) or *TNFRSF11A*	Suppression of osteoblast differentiation	Chen et al., 2018 [45]
	Integrin alpha-V (*ITGAV*)	Osteoblast differentiation promotion	Cheng et al., 2001 [46]
	Osteoclast-associated receptor (*OSCAR*)	Regulator of osteoclast differentiation	Barrow et al., 2011 [47]
	Activated leukocyte cell adhesion molecule (*ALCAM/CD166*)	Immature osteoblast marker, promotes osteoblast differentiation	Hooker et al., 2015 [48]
**Osteogenic intracellular markers**	Nuclear factor interleukin-3-regulated *(NFIL3*)	Transcriptional repressor in osteoblasts	Hariri et al., 2020 [49]
	Runt-related transcription factor 2 (*RUNX2*)	Essential for initial commitment of MSCs to the osteoblastic lineage	Camilleri et al., 2006 [50]
	T cell immune regulator 1 (*TCIRG1*)	Osteoclastogenesis regulation	Zhang et al., 2020 [51]
	WD repeat domain 5 protein (*WDR5*)	Critical for MSCs osteogenic differentiation	Zhu et al., 2016 [52]
	T-box 2 (*TBX2*)	Positive regulation of osteogenic differentiation	Govoni et al., 2009; Abrahams et al., 2010 [53,54]
	T-box 3 (*TBX3*)		
	Distal-less homeobox 5 (*DLX-5*)	Transcriptional regulation of osteoblast differentiation	Hassan et al., 2004 [55]
	Early B cell factor 2 (*EBF2*)	Inhibition of osteoblast differentiation	Kieslinger et al., 2005 [56]
**Osteogenic secreted markers**	Signal peptide, CUB, and EGF-like domain-containing protein 3 (*SCUBE3*)	Controlling growth, morphogenesis, and bone and teeth development	Lin et al., 2021 [57]
	Type 1 collagen A (*COL1A1*)	Early marker of osteoblast	Kannan et al., 2020 [58]
	Insulin-like growth factor binding protein-3 (*IGFBP-3*)	Osteoblasts differentiation suppression	Li et al., 2013 [59]
	Insulin-like growth factor binding protein-4 (*IGFBP-4*)	Osteoclastogenesis regulation	Maridas et al., 2017 [60]
	Secreted protein acidic and rich in cysteine (*SPARC*)	Regulation of bone remodeling and bone mass maintenance	Rosset et al., 2016 [61]
	SPARC-related modular calcium-binding protein 1 (*SMOC1*)	Increases the expression of osteoblast differentiation-related genes in BMSCs	Choi et al., 2010 [62]
	Acid phosphatase 5 (*ACP5*)	Promotion of odontoblast differentiation and mineralization during tooth development	Choi et al., 2016 [63]
	Biorientation of chromosomes in cell division1-like 1(*BOD1L1*)	Positive regulation of osteoblasts differentiation	Okamura et al., 2017 [64]; NCBI [65]
	Aggrecan (*ACAN*)	Bone tissue formation	Viti et al., 2016 [66]
	Tissue-nonspecific alkaline phosphatase (*ALPL*)	Essential for bone mineralization; osteoblast marker	Nakamura et al., 2020 [67]
	Biglycan (*BGN*)	Modulation of osteoblast differentiation	Parisuthiman et al., 2005 [68]
	Decorin (*DCN*)	Key marker of odontoblasts	Matsuura et al., 2001 [69]
	Fibronectin 1 (*FN1*)	Essential for osteoblast differentiation and mineralization	Globus et al., 1998 [70]
	Bone morphogenetic protein (*BMP2*)	Important in osteoblast differentiation	Yang et al., 2012 [71]

**Table 3 ijms-23-14420-t003:** Characteristics of total RNA extracted from the designated sample. The ‘No Bacteria’ type of treatment should be considered a control.

Sample	SCAP Source	Treatment	Concentration, pg/μL	Absorbance Ratio 260/280	RNA Integrity Number
1	Donor I	No Bacteria	4405	2.086	9.9
2	Donor I	*F. nucleatum* B *	3737	2.113	9.4
3	Donor I	*F. nucleatum* S **	3016	2.088	9.1
4	Donor I	*E. faecalis* B	3977	2.089	6.4
5	Donor I	*E. faecalis* S	3078	2.102	9.8
6	Donor II	No Bacteria	4791	2.097	9.8
7	Donor II	*F. nucleatum* B	6566	2.123	9.2
8	Donor II	*F. nucleatum* S	5288	2.096	10.0
9	Donor II	*E. faecalis* B	5208	2.087	8.0
10	Donor II	*E. faecalis* S	4465	2.080	9.5
11	Donor III	No Bacteria	4193	2.068	10.0
12	Donor III	*F. nucleatum* B	4929	2.058	8.7
13	Donor III	*F. nucleatum* S	4673	2.064	9.5
14	Donor III	*E. faecalis* B	4871	2.068	7.6
15	Donor III	*E. faecalis* S	4089	2.069	10.0

* B: bacteria, ** S: supernatants.

## Data Availability

The raw data supporting the conclusions of this article will be made available by the authors, without undue reservation.

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
