# Peer review of "Transcriptome Analysis Reveals Modulation of Human Stem Cells from the Apical Papilla by Species Associated with Dental Root Canal Infection"

_ijms, 2022, doi:10.3390/ijms232214420_

Round 1
Reviewer 1 Report
Reviewer
Article report.
Title: Transcriptome Analysis Reveals Modulation of Human Stem Cells from the Apical Papilla by Species Associated with Dental Root Canal Infection.
The objective of this article is to analyse modulation of human stem cells associated with tow bacteria F. nucleatum and E.faecalis in vitro . A better understanding of the behavior of stem cells with respect to bacterial infection can be an aid for new therapeutic approaches.
This study, based in part on tissue engineering, seeks to highlight the interactions between cells (SCAP) and two bacteria. But this research is only in its infancy. Also the usual reservations are to be made because this research only imperfectly reflects the reality of the pathology in vivo.
Thus several in vivo models have already been used. The design of an in vitro model that could sufficiently mimic the in vivo situation is still insufficiently developed. The difficulty in this context stems in part from the biological effects caused by viable or non-viable bacteria with respect to certain host cellular responses.
Pagination must be reorganize.
The article is from 2021? The pagination is aberrant: from 1 to 8 then from 1 to 3 and finally from 2 to 12/23. Line numbering begins at the discussion level through to the bibliography. This is an assembly that needs to be revamped.
Page 1
Abstract : it is preferable to clearly announce that the experimentation was carried out in vitro
Introduction:
Page 2. (lack of line numbering)
- More recent reference for (1) is suitable
- For reference (13). Generalizations should be avoided as this publication concerns only S. oralis and A. naeslundii (Petridis 2018)
- For reference (24). E. faecalis possesses many virulence factors, such as enterococcal surface protein (esp), gelatinase (gelE), aggregation substance (asa1), cytolysin B (cylB), and endocarditis-specific antigen A (efaA) gene, ArgR family transcription factor (ahrC), endocarditis and biofilmassociated pili (ebpA), enterococcal polysaccharide antigen (epal), epal and OG1RF_11715 (epaOX), and (p)ppGpp-synthetase/hydrolase (relA) genes.
For reference (26) relate to murine bone marrow (to be specified)
-
Page 3.
Results: Table 1. Why didn't you incubate F.nucleatum and E. faecalis with SCAP at the same time?
Page 7: Paragraphe 2.5 . Title . Here only F.nucleatum and F. faecalis and not all oral species.
Page 1 ?? : Table 2: Complete legend with: Genes associated with bone and dentin formation detected in SCAPs culture….
Discussion. start line numbering?
Page 1 ??? Line 12, ( precise healthy SCAP's)
Page 2 . Line 34. can you use the > sign instead of the arrows .
Page 2. Line 48. It should be noted that this result corresponds to a mouse model with non-healing wounds infected with E. faecalis. Depending on the dosage, colonization may be short-term and transient, while at high doses acute bacterial replication persists. In this case, the infiltration of immune cells leads to a delay in healing, despite the suppression of certain cytokines.
Page 2. Line 64. ref 76 . It should be noted that this publication (2005) is based on a model of bacterial infection of human intestinal epithelial cells. These authors extrapolate by demonstrating the effectiveness of error model-based microarray experiments and propose this as a general strategy for microarray-based screening of large collections of biological samples
Page 3. Line 94. Ref 83.. The results of this study concern arthropods and even invertebrates!. specify. Line 96. ref 45 (concerns the mouse).
Page 4. Line 138. Add your recent reference concerning the same experimental context: Zymovets, V.; Razghonova, Y.; Rakhimova, O.; Aripaka, K.; Manoharan, L.; Kelk, P.; Landstrom, M.; Romani Vestman, N. Combined Transcriptomic and Protein ArrayCytokineProfiling of Human StemCells from Dental Apical PapillaModulated by Oral Bacteria. Int.J.Mol. Sci. 2022, 23, 5098. https://doi.org/10.3390/ijms23095098. Line 140. (ref 94.95) Specify for epithelial cells.
Line 154. other limitations of in vitro research: only two bacteria are incriminated, for a short time healthy host cells of SCAP from only three teeth cannot reflect the situation of traumatized or infected teeth.
Page 5. Line 191. at what level of taxonomy did you identify F. nucleatum (species, subspecies?). Line 204. Which subspecies of F. nucleatum did you use as a reference in eHOMD? Number? Line 230. again why not include F. nucletum and E faecalis together with SCAPs?
Bibliographie
Page 8. Line 367 . ref 24. spelling of the authors font to be corrected of 24 ref .
Other considerations….concerning…
F. nucleatum.
F. nucleatum considered as a multifaceted commensal and opportunistic bacterium engages in various interactions with other microorganisms and human cells that range from beneficial to harmful in nature. At the biofilm level, F. nucleatum participates in the architecture and support of primary colonizers such as species (Streptococcus) then secondary colonizers such as Porphyromonas gingivalis and Aggregatibacter actinomycetemcomitans. F. nucleatum creates a microenvironment with reduced oxygen tension that protects P. gingivalis in the root canal.
Enterococcus faecalis is the most frequently isolated bacterial species from symptomatic root canal-treated teeth, with reported prevalence in up to 90% of cases .Endo, M.; Ferraz, C.; Zaia, A.A.; Almeida, J.F.A.; Gomes, B.P.F.A. Quantitative and qualitative analysis of microorganisms in
root-filled teeth with persistent infection: Monitoring of the endodontic retreatment. Eur. J. Dent. 2013, 7, 302–309.
E faecalis is often found in secondary or persistent cases due to its ability to survive in harsh environments with nutrient deprivation and high alkalinity despite the presence of intracanal medicament. The pathogenicity and difficulty of their eradication have been attributed to the ability of E. faecalis to form biofilms, which can be 1000-fold more resistant to antimicrobials than their planktonic counterparts.
The result presented in the article corresponds to an in vitro experiment. The transposition to the clinical context deserves some reservations that should be emphasized from the outset in the summary and at the end of the discussion. In particular, using healthy teeth for SCAP is in no way comparable to an infected or traumatized tooth. The article from this point of view represents only a rough approach to the complex in vivo ecosystem of the effects of oral bacteria on cell proliferation as well as on the differentiation capacity and immunomodulation of SCAPs of dental origin. The interest and the feasibility of the total elimination of the two opportunistic bacteria incriminated deserve to be further discussed on the one hand and on the other hand the interactions of these two bacteria with other species of the microbiota as well.
The text constitutes an assembly of different parts with many considerations. It would be better to go back to the 23 pages and go to the essential while keeping the benefit of the results. The bibliography dates from 2021 with many old references and needs to be completed. A recent reference by the same authors: Zymovets, V.; Razghonova, Y.; Rakhimova, O.; Aripaka, K.; Manoharan, L.; Kelk, P.; Landstrom, M.; Romani Vestman, N. Combined profiling of transcriptomic cytokines and protein networks of human stem cells from the dental apical papilla modulated by oral bacteria. Int. J.Mol. Science. 2022, to be included in the bibliography would update the article. Another recent reference from Tan, H.C.; Cheung, G.S.P. ; Chang, J.W.W. ; Zhang, C.; Lee, A.H.C. Enterococcus faecalis protects Porphyromonas gingivalis in a dual-species biofilm under oxic conditions. Microorganisms 2022, 10, 1729. https://doi.org/10.3390/microorganisms10091729, shows that E. faecalis and P. gingivalis have been frequently isolated from infected root canals and periodontal pockets by culture and identification techniques molecular.
Reviewer
Article report.
Title: Transcriptome Analysis Reveals Modulation of Human Stem Cells from the Apical Papilla by Species Associated with Dental Root Canal Infection.
The objective of this article is to analyse modulation of human stem cells associated with tow bacteria F. nucleatum and E.faecalis in vitro . A better understanding of the behavior of stem cells with respect to bacterial infection can be an aid for new therapeutic approaches.
This study, based in part on tissue engineering, seeks to highlight the interactions between cells (SCAP) and two bacteria. But this research is only in its infancy. Also the usual reservations are to be made because this research only imperfectly reflects the reality of the pathology in vivo.
Thus several in vivo models have already been used. The design of an in vitro model that could sufficiently mimic the in vivo situation is still insufficiently developed. The difficulty in this context stems in part from the biological effects caused by viable or non-viable bacteria with respect to certain host cellular responses.
Pagination must be reorganize.
The article is from 2021? The pagination is aberrant: from 1 to 8 then from 1 to 3 and finally from 2 to 12/23. Line numbering begins at the discussion level through to the bibliography. This is an assembly that needs to be revamped.
Page 1
Abstract : it is preferable to clearly announce that the experimentation was carried out in vitro
Introduction:
Page 2. (lack of line numbering)
- More recent reference for (1) is suitable
- For reference (13). Generalizations should be avoided as this publication concerns only S. oralis and A. naeslundii (Petridis 2018)
- For reference (24). E. faecalis possesses many virulence factors, such as enterococcal surface protein (esp), gelatinase (gelE), aggregation substance (asa1), cytolysin B (cylB), and endocarditis-specific antigen A (efaA) gene, ArgR family transcription factor (ahrC), endocarditis and biofilmassociated pili (ebpA), enterococcal polysaccharide antigen (epal), epal and OG1RF_11715 (epaOX), and (p)ppGpp-synthetase/hydrolase (relA) genes.
For reference (26) relate to murine bone marrow (to be specified)
-
Page 3.
Results: Table 1. Why didn't you incubate F.nucleatum and E. faecalis with SCAP at the same time?
Page 7: Paragraphe 2.5 . Title . Here only F.nucleatum and F. faecalis and not all oral species.
Page 1 ?? : Table 2: Complete legend with: Genes associated with bone and dentin formation detected in SCAPs culture….
Discussion. start line numbering?
Page 1 ??? Line 12, ( precise healthy SCAP's)
Page 2 . Line 34. can you use the > sign instead of the arrows .
Page 2. Line 48. It should be noted that this result corresponds to a mouse model with non-healing wounds infected with E. faecalis. Depending on the dosage, colonization may be short-term and transient, while at high doses acute bacterial replication persists. In this case, the infiltration of immune cells leads to a delay in healing, despite the suppression of certain cytokines.
Page 2. Line 64. ref 76 . It should be noted that this publication (2005) is based on a model of bacterial infection of human intestinal epithelial cells. These authors extrapolate by demonstrating the effectiveness of error model-based microarray experiments and propose this as a general strategy for microarray-based screening of large collections of biological samples
Page 3. Line 94. Ref 83.. The results of this study concern arthropods and even invertebrates!. specify. Line 96. ref 45 (concerns the mouse).
Page 4. Line 138. Add your recent reference concerning the same experimental context: Zymovets, V.; Razghonova, Y.; Rakhimova, O.; Aripaka, K.; Manoharan, L.; Kelk, P.; Landstrom, M.; Romani Vestman, N. Combined Transcriptomic and Protein ArrayCytokineProfiling of Human StemCells from Dental Apical PapillaModulated by Oral Bacteria. Int.J.Mol. Sci. 2022, 23, 5098. https://doi.org/10.3390/ijms23095098. Line 140. (ref 94.95) Specify for epithelial cells.
Line 154. other limitations of in vitro research: only two bacteria are incriminated, for a short time healthy host cells of SCAP from only three teeth cannot reflect the situation of traumatized or infected teeth.
Page 5. Line 191. at what level of taxonomy did you identify F. nucleatum (species, subspecies?). Line 204. Which subspecies of F. nucleatum did you use as a reference in eHOMD? Number? Line 230. again why not include F. nucletum and E faecalis together with SCAPs?
Bibliographie
Page 8. Line 367 . ref 24. spelling of the authors font to be corrected of 24 ref .
Other considerations….concerning…
F. nucleatum.
F. nucleatum considered as a multifaceted commensal and opportunistic bacterium engages in various interactions with other microorganisms and human cells that range from beneficial to harmful in nature. At the biofilm level, F. nucleatum participates in the architecture and support of primary colonizers such as species (Streptococcus) then secondary colonizers such as Porphyromonas gingivalis and Aggregatibacter actinomycetemcomitans. F. nucleatum creates a microenvironment with reduced oxygen tension that protects P. gingivalis in the root canal.
Enterococcus faecalis is the most frequently isolated bacterial species from symptomatic root canal-treated teeth, with reported prevalence in up to 90% of cases .Endo, M.; Ferraz, C.; Zaia, A.A.; Almeida, J.F.A.; Gomes, B.P.F.A. Quantitative and qualitative analysis of microorganisms in
root-filled teeth with persistent infection: Monitoring of the endodontic retreatment. Eur. J. Dent. 2013, 7, 302–309.
E faecalis is often found in secondary or persistent cases due to its ability to survive in harsh environments with nutrient deprivation and high alkalinity despite the presence of intracanal medicament. The pathogenicity and difficulty of their eradication have been attributed to the ability of E. faecalis to form biofilms, which can be 1000-fold more resistant to antimicrobials than their planktonic counterparts.
The result presented in the article corresponds to an in vitro experiment. The transposition to the clinical context deserves some reservations that should be emphasized from the outset in the summary and at the end of the discussion. In particular, using healthy teeth for SCAP is in no way comparable to an infected or traumatized tooth. The article from this point of view represents only a rough approach to the complex in vivo ecosystem of the effects of oral bacteria on cell proliferation as well as on the differentiation capacity and immunomodulation of SCAPs of dental origin. The interest and the feasibility of the total elimination of the two opportunistic bacteria incriminated deserve to be further discussed on the one hand and on the other hand the interactions of these two bacteria with other species of the microbiota as well.
The text constitutes an assembly of different parts with many considerations. It would be better to go back to the 23 pages and go to the essential while keeping the benefit of the results. The bibliography dates from 2021 with many old references and needs to be completed. A recent reference by the same authors: Zymovets, V.; Razghonova, Y.; Rakhimova, O.; Aripaka, K.; Manoharan, L.; Kelk, P.; Landstrom, M.; Romani Vestman, N. Combined profiling of transcriptomic cytokines and protein networks of human stem cells from the dental apical papilla modulated by oral bacteria. Int. J.Mol. Science. 2022, to be included in the bibliography would update the article. Another recent reference from Tan, H.C.; Cheung, G.S.P. ; Chang, J.W.W. ; Zhang, C.; Lee, A.H.C. Enterococcus faecalis protects Porphyromonas gingivalis in a dual-species biofilm under oxic conditions. Microorganisms 2022, 10, 1729. https://doi.org/10.3390/microorganisms10091729, shows that E. faecalis and P. gingivalis have been frequently isolated from infected root canals and periodontal pockets by culture and identification techniques molecular.
Reviewer
Article report.
Title: Transcriptome Analysis Reveals Modulation of Human Stem Cells from the Apical Papilla by Species Associated with Dental Root Canal Infection.
The objective of this article is to analyse modulation of human stem cells associated with tow bacteria F. nucleatum and E.faecalis in vitro . A better understanding of the behavior of stem cells with respect to bacterial infection can be an aid for new therapeutic approaches.
This study, based in part on tissue engineering, seeks to highlight the interactions between cells (SCAP) and two bacteria. But this research is only in its infancy. Also the usual reservations are to be made because this research only imperfectly reflects the reality of the pathology in vivo.
Thus several in vivo models have already been used. The design of an in vitro model that could sufficiently mimic the in vivo situation is still insufficiently developed. The difficulty in this context stems in part from the biological effects caused by viable or non-viable bacteria with respect to certain host cellular responses.
Pagination must be reorganize.
The article is from 2021? The pagination is aberrant: from 1 to 8 then from 1 to 3 and finally from 2 to 12/23. Line numbering begins at the discussion level through to the bibliography. This is an assembly that needs to be revamped.
Page 1
Abstract : it is preferable to clearly announce that the experimentation was carried out in vitro
Introduction:
Page 2. (lack of line numbering)
- More recent reference for (1) is suitable
- For reference (13). Generalizations should be avoided as this publication concerns only S. oralis and A. naeslundii (Petridis 2018)
- For reference (24). E. faecalis possesses many virulence factors, such as enterococcal surface protein (esp), gelatinase (gelE), aggregation substance (asa1), cytolysin B (cylB), and endocarditis-specific antigen A (efaA) gene, ArgR family transcription factor (ahrC), endocarditis and biofilmassociated pili (ebpA), enterococcal polysaccharide antigen (epal), epal and OG1RF_11715 (epaOX), and (p)ppGpp-synthetase/hydrolase (relA) genes.
For reference (26) relate to murine bone marrow (to be specified)
-
Page 3.
Results: Table 1. Why didn't you incubate F.nucleatum and E. faecalis with SCAP at the same time?
Page 7: Paragraphe 2.5 . Title . Here only F.nucleatum and F. faecalis and not all oral species.
Page 1 ?? : Table 2: Complete legend with: Genes associated with bone and dentin formation detected in SCAPs culture….
Discussion. start line numbering?
Page 1 ??? Line 12, ( precise healthy SCAP's)
Page 2 . Line 34. can you use the > sign instead of the arrows .
Page 2. Line 48. It should be noted that this result corresponds to a mouse model with non-healing wounds infected with E. faecalis. Depending on the dosage, colonization may be short-term and transient, while at high doses acute bacterial replication persists. In this case, the infiltration of immune cells leads to a delay in healing, despite the suppression of certain cytokines.
Page 2. Line 64. ref 76 . It should be noted that this publication (2005) is based on a model of bacterial infection of human intestinal epithelial cells. These authors extrapolate by demonstrating the effectiveness of error model-based microarray experiments and propose this as a general strategy for microarray-based screening of large collections of biological samples
Page 3. Line 94. Ref 83.. The results of this study concern arthropods and even invertebrates!. specify. Line 96. ref 45 (concerns the mouse).
Page 4. Line 138. Add your recent reference concerning the same experimental context: Zymovets, V.; Razghonova, Y.; Rakhimova, O.; Aripaka, K.; Manoharan, L.; Kelk, P.; Landstrom, M.; Romani Vestman, N. Combined Transcriptomic and Protein ArrayCytokineProfiling of Human StemCells from Dental Apical PapillaModulated by Oral Bacteria. Int.J.Mol. Sci. 2022, 23, 5098. https://doi.org/10.3390/ijms23095098. Line 140. (ref 94.95) Specify for epithelial cells.
Line 154. other limitations of in vitro research: only two bacteria are incriminated, for a short time healthy host cells of SCAP from only three teeth cannot reflect the situation of traumatized or infected teeth.
Page 5. Line 191. at what level of taxonomy did you identify F. nucleatum (species, subspecies?). Line 204. Which subspecies of F. nucleatum did you use as a reference in eHOMD? Number? Line 230. again why not include F. nucletum and E faecalis together with SCAPs?
Bibliographie
Page 8. Line 367 . ref 24. spelling of the authors font to be corrected of 24 ref .
Other considerations….concerning…
F. nucleatum.
F. nucleatum considered as a multifaceted commensal and opportunistic bacterium engages in various interactions with other microorganisms and human cells that range from beneficial to harmful in nature. At the biofilm level, F. nucleatum participates in the architecture and support of primary colonizers such as species (Streptococcus) then secondary colonizers such as Porphyromonas gingivalis and Aggregatibacter actinomycetemcomitans. F. nucleatum creates a microenvironment with reduced oxygen tension that protects P. gingivalis in the root canal.
Enterococcus faecalis is the most frequently isolated bacterial species from symptomatic root canal-treated teeth, with reported prevalence in up to 90% of cases .Endo, M.; Ferraz, C.; Zaia, A.A.; Almeida, J.F.A.; Gomes, B.P.F.A. Quantitative and qualitative analysis of microorganisms in
root-filled teeth with persistent infection: Monitoring of the endodontic retreatment. Eur. J. Dent. 2013, 7, 302–309.
E faecalis is often found in secondary or persistent cases due to its ability to survive in harsh environments with nutrient deprivation and high alkalinity despite the presence of intracanal medicament. The pathogenicity and difficulty of their eradication have been attributed to the ability of E. faecalis to form biofilms, which can be 1000-fold more resistant to antimicrobials than their planktonic counterparts.
The result presented in the article corresponds to an in vitro experiment. The transposition to the clinical context deserves some reservations that should be emphasized from the outset in the summary and at the end of the discussion. In particular, using healthy teeth for SCAP is in no way comparable to an infected or traumatized tooth. The article from this point of view represents only a rough approach to the complex in vivo ecosystem of the effects of oral bacteria on cell proliferation as well as on the differentiation capacity and immunomodulation of SCAPs of dental origin. The interest and the feasibility of the total elimination of the two opportunistic bacteria incriminated deserve to be further discussed on the one hand and on the other hand the interactions of these two bacteria with other species of the microbiota as well.
The text constitutes an assembly of different parts with many considerations. It would be better to go back to the 23 pages and go to the essential while keeping the benefit of the results. The bibliography dates from 2021 with many old references and needs to be completed. A recent reference by the same authors: Zymovets, V.; Razghonova, Y.; Rakhimova, O.; Aripaka, K.; Manoharan, L.; Kelk, P.; Landstrom, M.; Romani Vestman, N. Combined profiling of transcriptomic cytokines and protein networks of human stem cells from the dental apical papilla modulated by oral bacteria. Int. J.Mol. Science. 2022, to be included in the bibliography would update the article. Another recent reference from Tan, H.C.; Cheung, G.S.P. ; Chang, J.W.W. ; Zhang, C.; Lee, A.H.C. Enterococcus faecalis protects Porphyromonas gingivalis in a dual-species biofilm under oxic conditions. Microorganisms 2022, 10, 1729. https://doi.org/10.3390/microorganisms10091729, shows that E. faecalis and P. gingivalis have been frequently isolated from infected root canals and periodontal pockets by culture and identification techniques molecular.
Reviewer
Article report.
Title: Transcriptome Analysis Reveals Modulation of Human Stem Cells from the Apical Papilla by Species Associated with Dental Root Canal Infection.
The objective of this article is to analyse modulation of human stem cells associated with tow bacteria F. nucleatum and E.faecalis in vitro . A better understanding of the behavior of stem cells with respect to bacterial infection can be an aid for new therapeutic approaches.
This study, based in part on tissue engineering, seeks to highlight the interactions between cells (SCAP) and two bacteria. But this research is only in its infancy. Also the usual reservations are to be made because this research only imperfectly reflects the reality of the pathology in vivo.
Thus several in vivo models have already been used. The design of an in vitro model that could sufficiently mimic the in vivo situation is still insufficiently developed. The difficulty in this context stems in part from the biological effects caused by viable or non-viable bacteria with respect to certain host cellular responses.
Pagination must be reorganize.
The article is from 2021? The pagination is aberrant: from 1 to 8 then from 1 to 3 and finally from 2 to 12/23. Line numbering begins at the discussion level through to the bibliography. This is an assembly that needs to be revamped.
Page 1
Abstract : it is preferable to clearly announce that the experimentation was carried out in vitro
Introduction:
Page 2. (lack of line numbering)
- More recent reference for (1) is suitable
- For reference (13). Generalizations should be avoided as this publication concerns only S. oralis and A. naeslundii (Petridis 2018)
- For reference (24). E. faecalis possesses many virulence factors, such as enterococcal surface protein (esp), gelatinase (gelE), aggregation substance (asa1), cytolysin B (cylB), and endocarditis-specific antigen A (efaA) gene, ArgR family transcription factor (ahrC), endocarditis and biofilmassociated pili (ebpA), enterococcal polysaccharide antigen (epal), epal and OG1RF_11715 (epaOX), and (p)ppGpp-synthetase/hydrolase (relA) genes.
For reference (26) relate to murine bone marrow (to be specified)
-
Page 3.
Results: Table 1. Why didn't you incubate F.nucleatum and E. faecalis with SCAP at the same time?
Page 7: Paragraphe 2.5 . Title . Here only F.nucleatum and F. faecalis and not all oral species.
Page 1 ?? : Table 2: Complete legend with: Genes associated with bone and dentin formation detected in SCAPs culture….
Discussion. start line numbering?
Page 1 ??? Line 12, ( precise healthy SCAP's)
Page 2 . Line 34. can you use the > sign instead of the arrows .
Page 2. Line 48. It should be noted that this result corresponds to a mouse model with non-healing wounds infected with E. faecalis. Depending on the dosage, colonization may be short-term and transient, while at high doses acute bacterial replication persists. In this case, the infiltration of immune cells leads to a delay in healing, despite the suppression of certain cytokines.
Page 2. Line 64. ref 76 . It should be noted that this publication (2005) is based on a model of bacterial infection of human intestinal epithelial cells. These authors extrapolate by demonstrating the effectiveness of error model-based microarray experiments and propose this as a general strategy for microarray-based screening of large collections of biological samples
Page 3. Line 94. Ref 83.. The results of this study concern arthropods and even invertebrates!. specify. Line 96. ref 45 (concerns the mouse).
Page 4. Line 138. Add your recent reference concerning the same experimental context: Zymovets, V.; Razghonova, Y.; Rakhimova, O.; Aripaka, K.; Manoharan, L.; Kelk, P.; Landstrom, M.; Romani Vestman, N. Combined Transcriptomic and Protein ArrayCytokineProfiling of Human StemCells from Dental Apical PapillaModulated by Oral Bacteria. Int.J.Mol. Sci. 2022, 23, 5098. https://doi.org/10.3390/ijms23095098. Line 140. (ref 94.95) Specify for epithelial cells.
Line 154. other limitations of in vitro research: only two bacteria are incriminated, for a short time healthy host cells of SCAP from only three teeth cannot reflect the situation of traumatized or infected teeth.
Page 5. Line 191. at what level of taxonomy did you identify F. nucleatum (species, subspecies?). Line 204. Which subspecies of F. nucleatum did you use as a reference in eHOMD? Number? Line 230. again why not include F. nucletum and E faecalis together with SCAPs?
Bibliographie
Page 8. Line 367 . ref 24. spelling of the authors font to be corrected of 24 ref .
Other considerations….concerning…
F. nucleatum.
F. nucleatum considered as a multifaceted commensal and opportunistic bacterium engages in various interactions with other microorganisms and human cells that range from beneficial to harmful in nature. At the biofilm level, F. nucleatum participates in the architecture and support of primary colonizers such as species (Streptococcus) then secondary colonizers such as Porphyromonas gingivalis and Aggregatibacter actinomycetemcomitans. F. nucleatum creates a microenvironment with reduced oxygen tension that protects P. gingivalis in the root canal.
Enterococcus faecalis is the most frequently isolated bacterial species from symptomatic root canal-treated teeth, with reported prevalence in up to 90% of cases .Endo, M.; Ferraz, C.; Zaia, A.A.; Almeida, J.F.A.; Gomes, B.P.F.A. Quantitative and qualitative analysis of microorganisms in
root-filled teeth with persistent infection: Monitoring of the endodontic retreatment. Eur. J. Dent. 2013, 7, 302–309.
E faecalis is often found in secondary or persistent cases due to its ability to survive in harsh environments with nutrient deprivation and high alkalinity despite the presence of intracanal medicament. The pathogenicity and difficulty of their eradication have been attributed to the ability of E. faecalis to form biofilms, which can be 1000-fold more resistant to antimicrobials than their planktonic counterparts.
The result presented in the article corresponds to an in vitro experiment. The transposition to the clinical context deserves some reservations that should be emphasized from the outset in the summary and at the end of the discussion. In particular, using healthy teeth for SCAP is in no way comparable to an infected or traumatized tooth. The article from this point of view represents only a rough approach to the complex in vivo ecosystem of the effects of oral bacteria on cell proliferation as well as on the differentiation capacity and immunomodulation of SCAPs of dental origin. The interest and the feasibility of the total elimination of the two opportunistic bacteria incriminated deserve to be further discussed on the one hand and on the other hand the interactions of these two bacteria with other species of the microbiota as well.
The text constitutes an assembly of different parts with many considerations. It would be better to go back to the 23 pages and go to the essential while keeping the benefit of the results. The bibliography dates from 2021 with many old references and needs to be completed. A recent reference by the same authors: Zymovets, V.; Razghonova, Y.; Rakhimova, O.; Aripaka, K.; Manoharan, L.; Kelk, P.; Landstrom, M.; Romani Vestman, N. Combined profiling of transcriptomic cytokines and protein networks of human stem cells from the dental apical papilla modulated by oral bacteria. Int. J.Mol. Science. 2022, to be included in the bibliography would update the article. Another recent reference from Tan, H.C.; Cheung, G.S.P. ; Chang, J.W.W. ; Zhang, C.; Lee, A.H.C. Enterococcus faecalis protects Porphyromonas gingivalis in a dual-species biofilm under oxic conditions. Microorganisms 2022, 10, 1729. https://doi.org/10.3390/microorganisms10091729, shows that E. faecalis and P. gingivalis have been frequently isolated from infected root canals and periodontal pockets by culture and identification techniques molecular.
Reviewer
Article report.
Title: Transcriptome Analysis Reveals Modulation of Human Stem Cells from the Apical Papilla by Species Associated with Dental Root Canal Infection.
The objective of this article is to analyse modulation of human stem cells associated with tow bacteria F. nucleatum and E.faecalis in vitro . A better understanding of the behavior of stem cells with respect to bacterial infection can be an aid for new therapeutic approaches.
This study, based in part on tissue engineering, seeks to highlight the interactions between cells (SCAP) and two bacteria. But this research is only in its infancy. Also the usual reservations are to be made because this research only imperfectly reflects the reality of the pathology in vivo.
Thus several in vivo models have already been used. The design of an in vitro model that could sufficiently mimic the in vivo situation is still insufficiently developed. The difficulty in this context stems in part from the biological effects caused by viable or non-viable bacteria with respect to certain host cellular responses.
Pagination must be reorganize.
The article is from 2021? The pagination is aberrant: from 1 to 8 then from 1 to 3 and finally from 2 to 12/23. Line numbering begins at the discussion level through to the bibliography. This is an assembly that needs to be revamped.
Page 1
Abstract : it is preferable to clearly announce that the experimentation was carried out in vitro
Introduction:
Page 2. (lack of line numbering)
- More recent reference for (1) is suitable
- For reference (13). Generalizations should be avoided as this publication concerns only S. oralis and A. naeslundii (Petridis 2018)
- For reference (24). E. faecalis possesses many virulence factors, such as enterococcal surface protein (esp), gelatinase (gelE), aggregation substance (asa1), cytolysin B (cylB), and endocarditis-specific antigen A (efaA) gene, ArgR family transcription factor (ahrC), endocarditis and biofilmassociated pili (ebpA), enterococcal polysaccharide antigen (epal), epal and OG1RF_11715 (epaOX), and (p)ppGpp-synthetase/hydrolase (relA) genes.
For reference (26) relate to murine bone marrow (to be specified)
-
Page 3.
Results: Table 1. Why didn't you incubate F.nucleatum and E. faecalis with SCAP at the same time?
Page 7: Paragraphe 2.5 . Title . Here only F.nucleatum and F. faecalis and not all oral species.
Page 1 ?? : Table 2: Complete legend with: Genes associated with bone and dentin formation detected in SCAPs culture….
Discussion. start line numbering?
Page 1 ??? Line 12, ( precise healthy SCAP's)
Page 2 . Line 34. can you use the > sign instead of the arrows .
Page 2. Line 48. It should be noted that this result corresponds to a mouse model with non-healing wounds infected with E. faecalis. Depending on the dosage, colonization may be short-term and transient, while at high doses acute bacterial replication persists. In this case, the infiltration of immune cells leads to a delay in healing, despite the suppression of certain cytokines.
Page 2. Line 64. ref 76 . It should be noted that this publication (2005) is based on a model of bacterial infection of human intestinal epithelial cells. These authors extrapolate by demonstrating the effectiveness of error model-based microarray experiments and propose this as a general strategy for microarray-based screening of large collections of biological samples
Page 3. Line 94. Ref 83.. The results of this study concern arthropods and even invertebrates!. specify. Line 96. ref 45 (concerns the mouse).
Page 4. Line 138. Add your recent reference concerning the same experimental context: Zymovets, V.; Razghonova, Y.; Rakhimova, O.; Aripaka, K.; Manoharan, L.; Kelk, P.; Landstrom, M.; Romani Vestman, N. Combined Transcriptomic and Protein ArrayCytokineProfiling of Human StemCells from Dental Apical PapillaModulated by Oral Bacteria. Int.J.Mol. Sci. 2022, 23, 5098. https://doi.org/10.3390/ijms23095098. Line 140. (ref 94.95) Specify for epithelial cells.
Line 154. other limitations of in vitro research: only two bacteria are incriminated, for a short time healthy host cells of SCAP from only three teeth cannot reflect the situation of traumatized or infected teeth.
Page 5. Line 191. at what level of taxonomy did you identify F. nucleatum (species, subspecies?). Line 204. Which subspecies of F. nucleatum did you use as a reference in eHOMD? Number? Line 230. again why not include F. nucletum and E faecalis together with SCAPs?
Bibliographie
Page 8. Line 367 . ref 24. spelling of the authors font to be corrected of 24 ref .
Other considerations….concerning…
F. nucleatum.
F. nucleatum considered as a multifaceted commensal and opportunistic bacterium engages in various interactions with other microorganisms and human cells that range from beneficial to harmful in nature. At the biofilm level, F. nucleatum participates in the architecture and support of primary colonizers such as species (Streptococcus) then secondary colonizers such as Porphyromonas gingivalis and Aggregatibacter actinomycetemcomitans. F. nucleatum creates a microenvironment with reduced oxygen tension that protects P. gingivalis in the root canal.
Enterococcus faecalis is the most frequently isolated bacterial species from symptomatic root canal-treated teeth, with reported prevalence in up to 90% of cases .Endo, M.; Ferraz, C.; Zaia, A.A.; Almeida, J.F.A.; Gomes, B.P.F.A. Quantitative and qualitative analysis of microorganisms in
root-filled teeth with persistent infection: Monitoring of the endodontic retreatment. Eur. J. Dent. 2013, 7, 302–309.
E faecalis is often found in secondary or persistent cases due to its ability to survive in harsh environments with nutrient deprivation and high alkalinity despite the presence of intracanal medicament. The pathogenicity and difficulty of their eradication have been attributed to the ability of E. faecalis to form biofilms, which can be 1000-fold more resistant to antimicrobials than their planktonic counterparts.
The result presented in the article corresponds to an in vitro experiment. The transposition to the clinical context deserves some reservations that should be emphasized from the outset in the summary and at the end of the discussion. In particular, using healthy teeth for SCAP is in no way comparable to an infected or traumatized tooth. The article from this point of view represents only a rough approach to the complex in vivo ecosystem of the effects of oral bacteria on cell proliferation as well as on the differentiation capacity and immunomodulation of SCAPs of dental origin. The interest and the feasibility of the total elimination of the two opportunistic bacteria incriminated deserve to be further discussed on the one hand and on the other hand the interactions of these two bacteria with other species of the microbiota as well.
The text constitutes an assembly of different parts with many considerations. It would be better to go back to the 23 pages and go to the essential while keeping the benefit of the results. The bibliography dates from 2021 with many old references and needs to be completed. A recent reference by the same authors: Zymovets, V.; Razghonova, Y.; Rakhimova, O.; Aripaka, K.; Manoharan, L.; Kelk, P.; Landstrom, M.; Romani Vestman, N. Combined profiling of transcriptomic cytokines and protein networks of human stem cells from the dental apical papilla modulated by oral bacteria. Int. J.Mol. Science. 2022, to be included in the bibliography would update the article. Another recent reference from Tan, H.C.; Cheung, G.S.P. ; Chang, J.W.W. ; Zhang, C.; Lee, A.H.C. Enterococcus faecalis protects Porphyromonas gingivalis in a dual-species biofilm under oxic conditions. Microorganisms 2022, 10, 1729. https://doi.org/10.3390/microorganisms10091729, shows that E. faecalis and P. gingivalis have been frequently isolated from infected root canals and periodontal pockets by culture and identification techniques molecular.
Author Response
Please, see the attachment.

Reviewer 2 Report
In this study, the authors revealed that species associated with dental root canal infection including F. nucleatum, E. faecalis, and their metabolites negatively regulated the dentinogenesis and osteogenesis of stem cells from the apical papilla (SCAP). The research is interesting. However, there are not enough experimental results to support this conclusion as detailed in the follows.
1. In the Materials and Methods section of the paper, the author mentioned that the authenticity of SCAP was confirmed by flow cytometry (FCM). But no data were presented so please provide the relating results of FCM. Similarly, please provide the relating PCR results of identification after F. nucleatum and E. faecalis were clinical isolated.
2. The transcriptome results of RNA-seq were analyzed in detail and thoroughly, but the authors inferred that F. nucleatum and E. faecalis regulated the dentinogenesis and osteogenesis of SCAP merely based on RNA-seq transcriptomic analysis, without further experimental verification (such as PCR etc.) of dentinogenic and osteogenic genes revealed by the RNA-seq transcriptomic analysis, which lacked sufficient evidence to support this conclusion.
3. SCAPs were co-cultured directly with F. nucleatum, E. faecalis and their respective supernatants and these two bacterial species and their respective supernatant exhibited different effects on SCAP in this study. Why both direct bacterial co-culture and bacterial supernatant co-culture with SCAP chosen? Besides, please discuss the potential mechanisms of the effect of bacterial species or their metabolites on the dentinogenesis and osteogenesis SCAP.
4. The experimental groups in this study are five groups, but there were only four groups in Venn diagram demonstrated in Figure 2.
Author Response
Please, see the attachment.

Round 2
Reviewer 1 Report
Regarding Fusobacterium nucleatum used in your experiment, do you have more details on the subspecies?
Reviewer 2 Report
The required revisions were made according to my comments to the Author. Thank you for your careful revision.